# Low bone mineral density in HIV-positive young Italians and migrants

**Antonio Cascio[1], Claudia Colomba[1], Paola Di Carlo[1], Nicola Serra[2], Giuseppe Lo Re[3], Angelo Gambino[3], Antonio Lo Casto[4], Giuseppe Guglielmi[5,6], Nicola Veronese[7], Roberto Lagalla[3], Consolato Sergi[8,9]***

**1** Department of Health Promotion and Child Health, University Hospital, Palermo, Italy, **2** Department of Molecular Medicine and Medical Biotechnology, University Federico II of Naples, Naples, Italy, **3** Department of Biopathology and Biotechnologies (DiBiMed), University Hospital 'Paolo Giaccone', Palermo, Italy, **4** Department of Biomedicine, Neuroscience and Advanced Diagnostics (BIND), University of Palermo, Palermo, Italy, **5** Department of Radiology, University of Foggia, Foggia, Italy, **6** Department of Radiology, Scientific Institute "Casa Sollievo della Sofferenza" Hospital, San Giovanni Rotondo, Foggia, Italy, **7** National Research Council, Neuroscience Institute, Aging Branch, Padova, Italy, **8** Department of Lab. Medicine and Pathology, University of Alberta, Edmonton, AB, Canada, **9** Stollery Children's Hospital, Edmonton, AB, Canada

* sergi@ualberta.ca

**Data Availability Statement:** Data contain potentially identifying or sensitive patient information and there are ethical and legal restrictions on sharing a de-identified data set. Data are stored at the University of Palermo, "P.

## Abstract

### Background

Human immunodeficiency virus (HIV) infected individuals may have osteoporosis. We aimed to evaluate the bone mineral density (BMD) in naïve antiretroviral (ARV) treated HIV positive patients comparing native Italian group (ItG) to a Migrants group (MiG) upon arrival in Italy.

### Methods

We conducted a cross-sectional study on 83 HIV patients less than 50 years old. We used the dual-energy X-ray absorptiometry (DXA) within six months from the HIV diagnosis. Participants were categorized as having low BMD if the femoral neck or total lumbar spine Z-score was– 2 or less.

### Results

MiG showed low BMD more often than ItG (37.5% vs.13.6%), especially for the female gender (16.7% vs. 0.0%). A low CD4 rate (<200 cells/µl) was most often detected in MiG than ItG. In particular, we found most often male Italians with abnormal CD4 than male migrants (67.8% vs. 33.3%) and vice versa for females (30.5% vs. 66.7%). We found an abnormal bone mineral density at the lumbar site. Low BMD at the lumbar site was more frequently observed in female migrants than female Italians. Both male and female migrants had a Z-score value significantly lower than male and female Italians, respectively. By logistic regression low vitamin-D level was positively correlated to low BMD in ItG only. All data were verified and validated using a triple code identifier.

Giaccone" Policlinic Institutional Electronic Data Repository (Via del Vespro, 129, 90127 Palermo PA, Italy). Access is restricted by application of credentials (username and password). The Ethics Committee can grant access to researchers who meet the criteria for access to confidential data (Bioethics Committee - Azienda Ospedaliera Universitaria Policlinico Paolo Giaccone, Via del Vespro, 129, Palermo IT 90127, Italy, Phone+39-91-6551111, Fax:+39-91-6555611, Email: bioetica@policlinico.pa.it).

**Funding:** This study was funded by an intramural grant of the University of Palermo, Italy (Grant ID #: D40001D05-0002).

**Competing interests:** The authors have declared that no competing interests exist.

## Conclusions

Both DXA and vitamin-D evaluation should be offered after the diagnosis of HIV infection. Lumbar site low BMD is an initial condition of bone loss in HIV young patients, especially in female migrants. Vitamin D levels and supplementation may be considered after HIV diagnosis independently of age to improve bone health.

## Highlights

This study evaluates the frequency of bone mineral density in HIV positive patients naive to antiretroviral therapy. It compares the density of the native Italian population with that of HIV Migrants upon arrival in Italy. The results show that HIV positive migrants, even if younger than 50 years of age, are at risk for osteoporosis, especially if they are female.

## Introduction

Osteoporosis is a process characterized by increased bone resorption without reciprocal bone apposition. The decreased bone mass leads to an increased risk of bone fractures, which are particularly as the subject ages [1, 2]. Currently, lifestyle factors, including alcohol, diet, hormones, physical activity, and smoking, are influencing the bone mass, and osteoporosis is affecting younger people than before [2–4]. The incidence of osteoporosis and osteoporosis-related fractures varies across the European Union/European Economic Area (EU/EEA), and especially in the Mediterranean area, it is increasing, as it has been evidenced in the literature with a clear link to nutrition [1, 2].

Among human immunodeficiency virus (HIV)-related comorbidities, the bone disease has emerged as a significant question in this chronically infected population [3–8]. Several studies suggest that fracture rates are higher in communities with HIV than among matched uninfected controls despite anti-retroviral therapy (ART) and gender [9–11]. The causes of a low bone mineral density (BMD) could be different in HIV-infected patients by considering the time of HIV infection [12]. Previous studies conducted in our geographical area established that the prevalence of osteoporosis was significantly higher in HIV- infected than in uninfected subjects, which mirrors a result similar to previous meta-analyses [12–15]. The prevalence of osteopenia and osteoporosis in HIV mono-infected patients in our geographical area was about 44.9%, and 20.9% in comparison with 18% reported in a healthy Italian population [16].

A recent analysis of bone-healthy in the immigrant population conducted in Sweden showed that women had low BMD for age according to the American and African–American referents [17–19]. Moreover, a study conducted in German-Turkish immigrants osteopenia was diagnosed in 32% and osteoporosis in 8% of young migrants [20]. In the migrant setting, it has been hypothesized that BMD is intriguingly associated with lifestyle, body mass index (BMI), and vitamin D levels. Recently, low level of vitamin D were also reported in Somali migrant women in Sweden and Refugees in Canada using the Calgary Refugee Health Program, an urban family practice that serves newly arrived refugees in Calgary, Alberta [21–23].

Although the number of migrants and refugees crossing the Mediterranean Sea has decreased in 2018–19 and 2020, during the COVID-19 pandemic, this number has been unprecedented in the last ten years compared with the past twenty or thirty years [5–7]. Topical Italian data showed that the new diagnosis of HIV infection in 72% of migrants was late and less than six months before developing AIDS with half of the subjects coming from Sub-

Saharan Africa (SSA, 59.4%) and with an increasing percentage of fertile females [22–26]. Further, the incidence of BMD loss and related fractures attributed to a specific eating pattern has also been targeted for guidelines and suggestions within the EU/EEA [8].

Our study aims to emphasize the burden of bone health in naïve ARV HIV positive patients and compare the bone density of the native Italian population group (ItG) with that of HIV Migrants (MiG) upon arrival in Italy.

## Materials and methods

### Study design and setting

This investigation is a retrospective cross-sectional study. We gathered data from 83 ARV naïve subjects consecutively admitted between January 2010 and May 2015 at the Sicilian AIDS Center of the University of Palermo, Italy. We retrospectively analyzed all patients who underwent dual-energy X-ray absorptiometry (DXA) within six months from HIV diagnosis independent of their BMI. In this study, we excluded all patients with the treatment of steroid-induced bone loss and subjects with active tuberculosis.

DXA measures BMD utilizing two X-ray beams with different energy levels, which are aimed at the patient's bones (see below). Of the 83 subjects, 59 were Italian, and 24 were migrant patients living in Italy for less than twelve months. **Table 1** shows the characteristics

**Table 1. Characteristics of the 83 HIV study participants.**

| Parameters | % (Number) |
| --- | --- |
| Patients | 83 |
| Italian | 71.1% (59) |
| male | 49.4% (41) |
| female | 21.7% (18) |
| Migrant | 28.9% (24) |
| male | 9.6% (8) |
| female | 19.3% (16) |
| Age, in years | |
| mean ±SD | 44.2 ±4.9 |
| Range | 30–50 |
| % patients with abnormal BMI score | 49.4% (41) |
| CD4 count $\leq$ 500 (cells/μl) | 98.8% (82) |
| CD4 count(cells/μl) | |
| <200 | 55.4% (46) |
| 200–349 | 41.0% (34) |
| $\geq$350 | 3.6% (3) |
| % patients with a low level of 25-Hydroxy-vitamin D | 100% (83) |
| % patients with AIDS | 56.6% (47) |
| % patients Mono-infected (HIV) | 81.9% (68) |
| % patients Co-infected (HIV and HCV) | 18.1% (15) |
| % patients with low BMD (ItG+MiG) | 20.1% (17 = 8+9) |
| % patients with Previous fractures | 9.6% (8) |

BMI = body mass index: normal range [18, 5–24, 9]

CD4 count: normal range [500–1500] (cells/μl)

Vitamin D: normal range [30–100] (ng/mL)[a]

Low BMD = Low Bone Mineral Density (Z-score $\leq$-2).

of enrolled patients. We collected the medical records and entered into an anonymous database, as previously reported [8, 9]. CD4+ T-cell count and plasma HIV-RNA levels were assessed as previously reported [9].

25-hydroxy-vitamin D (25(OH)D) was assayed as previously reported [21, 22, 25]. We considered a serum 25(OH)D concentration of 30 ng/mL as a threshold value for identifying low levels of vitamin D, as previously reported (standard value of our laboratory: 30–80 ng/mL) [13].

All subjects gave their informed consent for inclusion before they started participating in the study, which was conducted by the Declaration of Helsinki. The Ethics Committee of the University of Palermo approved the study protocol. Each participant was free to decline or withdraw from the study at any time, and this was explained in their original language or a language that the migrant knew.

## Bone mineral density assessment

BMD was assessed by DXA, using a QDR Discovery Hologic DXA in the femoral neck and DXA in the lumbar spine by total body DXA. For each scan, BMD, Z-scores were recorded as previously reported [9–11]. DXA measurements were performed in the femur (femoral neck and/or total hip) and lumbar spine in each patient. Since the age of our patients ranged from 30 to 50 years, the use of Z-scores (defined as an individuals' BMD in comparison to age-matched normal individuals) was used for all the analyses, according to World Health Organisation (WHO) recommendation [27]. Participants were categorized as having low BMD if the femoral neck or total lumbar spine Z-score was– 2 or less.

## Sample size estimation

To individualize sample sizes statistically significant in this study, we considered a Bernoulli sampling for both Italian and Migrant groups [28].

For ItG, the minimum sample size for this study was estimated equal to 41 patients affected by low BMD. It was obtained considering a statistical z-score at 95%, an error $\varepsilon$ = 15% and hypothesizing a prevalence $\pi$, about 60% according to the studies of Cavalli et al. [16] and Tomažič et al. [29]. In this case, we estimated a prevalence range equal to 45%-75%.

For MiG, the minimum sample size for this study was estimated equal to 22 patients affected by low BMD. It was obtained considering a z-score at 95%, an error $\varepsilon$ = 20%, and hypothesizing a prevalence $\pi$ about 35%, according to Varenna et al. [30]. In this way we estimated a prevalence range equal to 15%-55%. In this case, we considered an error $\varepsilon$ greater to an error in ItG because of the amount of information that could be gathered in migrants in Italy. Also, the sample size for ItG and MiG were increased to 59 and 24 patients, respectively considering the possibility of unexpected events and, consequently, the likelihood of patients' data loss.

## Statistical analysis

The statistical analysis was performed by MATLAB statistical toolbox version 2008 (MathWorks, Natick, MA, USA). Data are presented as number and percentage for categorical variables. Numerical data are expressed as the mean ± standard deviation (SD) or median and confidence interval at 95% (CI). The $\chi^2$ test and Fisher's exact tests were performed to evaluate significant differences of proportions or percentages between two groups. The Fisher's exact test was used where the $\chi^2$ test was not appropriate. The Mann-Whitney test was used to test the difference between two independent samples (ItG and MiG group). It was the alternative for a $t$-test dealing with independent samples when the distribution of the samples is not

normal. Normal distributed independent samples were studied using the D'Agostino-Pearson test. Linear correlation analysis was also performed, and the test on Pearson's linear correlation coefficient *r* was performed with the t-Student test under the null hypothesis of Pearson's linear correlation coefficient *r* = 0. The logistic regression was performed to analyze the relationship between Low BMD (dichotomous variable) and the independent variables: Gender (dichotomous–M = 1, F = 0), Previous fractures (dichotomous–yes = 1, no = 0), BMI (continuous), CD4 cells (continuous), and 25-Hydroxy-vitamin D (continuous). Finally, all tests with p-value < 0.05 were considered significant.

## Results

The 83 patients, composed by 59.4% males and 40.6% females, with ages into range 30–50, mean 44.2 years old and standard deviation (S.D.) equal to 4.9 years old, was subdivided into two groups (Table 1); the ItG (Italian group), composed of 59 patients, with ages into range 30–49, mean 43.7 years old and standard deviation (S.D.) equal to 4.8 years old. The MiG (migrant group), composed of 24 patients, with ages into range 32–50, mean 45.4 years old, and standard deviation (S.D.) equal to 4.8 years old. In detail, 22/24 (92%) migrants came from Africa and 2/24 (8%) from Asia.

In **Table 2**, we report the differences between ItG and MiG group and statistical tests. Low BMD was significantly more frequent in MiG group in comparison to ItG group (37.5% > 13.6%, *p* = 0.0324), particularly it was significantly more frequently found in migrant females than in Italian females (16.7% > 0.0%, *p* = 0.0058). Significant differences were seen in low-BMD values. In fact, both female and male Italians had a measure of BMD significantly greater than migrant females and males respectively (median: -1.0 > -2.3, *p* = 0.0272; 0.05 > -0.8, *p* = 0.0291). In addition, we found major presence of migrant females with abnormal lumbar value in comparison to Italian females (16.7% > 0.0%, *p* = 0.0058).

Conversely, no significant differences between ItG and MiG were found considering the femoral BMD values.

In examining the gender in mono-infected HIV patients, the percentages of HIV patients were significantly greater in Italian males in comparison to migrant males (57.6% > 25%, *p* = 0.007). On the other hand, HIV was most frequently observed in female migrants in comparison to female Italians (58.3% > 23.7%, *p* < 0.003). For co-infected patients, there were no significant differences in gender between ItG and MiG. Also, for AIDS patients there was a significant difference between ItG and MiG (45.8% < 83.3%, *p* = 0.0017). In particular, AIDS was most frequent in female Migrants in comparison to female Italians (8.3%>8.5%, *p*<0.0001). We found a higher percentage of male Italians with abnormal values of CD4 in comparison to male migrants (67.8% > 33.3%, *p* = 0.0039), and vice versa for the female gender (30.5%< 66.7%, *p* = 0.0024). With reference of CD4 values (median), there was a significant difference between ItG and MiG (210 vs 61, p = 0.0003), analogous for male and female (CD4 median: 200 vs 37.5, *p* = 0.0264; CD4 median: 265 vs 72.5, *p* = 0.0016). Low CD4 values (<200 cells/μl) in peripheral blood were more frequently found in Migrants than Italian patients (44.1% < 83.3%, *p* = 0.0012), while CD4 values into range 200–349 cells/μl were more frequently found in ItG in comparison to MiG patients (50.8% > 16.7%, *p* = 0.0043). There were no significant differences between ItG and MiG concerning high CD4 values (≥ 350 cells/μl). Regarding the BMI, male Italians with abnormal values of BMI were more often seen than male Migrants (33.9% > 12.5%, *p* = 0.0483). Further, BMI values of female Migrants were greater than BMI values of female Italians (median: 21 > 18.4, *p* = 0.0079).

In migrant patients, 25-OH-vitamin D mean values were significantly lower than in Italian patients (14.8 > 11.8 ng/ml, *p* = 0.0466). Also, male Italians had mean value significantly

**Table 2. Differences between ItG and MiG groups.**

| Parameters | ItG Mean ± SD or percentage | MiG Mean ± SD or percentage | ItG vs. MiG p-value |
|---|---|---|---|
| Nr. Patients | 59 | 24 | |
| Age | 46 [43–47] | 47.5 [43–49] | 0.0423* (MW) |
| Male | 44 [41–47] | 44.5 [37.8–49.2] | 0.48 (MW) |
| Female | 47 [44.8–47] | 47.5 [44.9–49.4] | 0.18 (MW) |
| Gender | | | |
| Male | 69.5% (41/59) | 33.3%(8/24) | 0.0025 * (C) |
| Female | 30.5% (18/59) | 66.7% (16/24) | 0.0025 * (C) |
| Low BMD | 13.6% (8/59) | 37.5% (9/24) | 0.0324 * (F) |
| Male | 13.6% (8/59) | 20.8% (5/24) | 0.51 (F) |
| Female | 0.0% (0/59) | 16.7% (4/24) | 0.0058 * (F) |
| Mono-infected (HIV) | 81.4% (48/59) | 83.3% (20/24) | 0.83 (C) |
| Male | 57.6% (34/59) | 25% (6/24) | 0.007* (C) |
| Female | 23.7% (14/59) | 58.3% (14/24) | 0.003* (C) |
| Co-infected (HIV and HCV) | 18.6% (11/59) | 16.7% (4/24) | 1.00 (F) |
| Male | 11.9% (7/59) | 8.3% (2/24) | 1.00 (F) |
| Female | 6.8% (4/59) | 8.3% (2/24) | 1.00 (F) |
| AIDS | 45.8% (27/59) | 83.3% (20/24) | 0.0017* (C) |
| Male | 37.3% (22/59) | 25.0% (6/24) | 0.28 (C) |
| Female | 8.5% (5/59) | 58.3% (14/24) | <0.0001 * (C) |
| CD4 count (cells/μl) | | | |
| <200 | 44.1% (26/59) | 83.3% (20/24) | 0.0012* (C) |
| 200–349 | 50.8% (30/59) | 16.7% (4/24) | 0.0043* (C) |
| ≥350 | 5.1% (3/59) | 0.0% (0/24) | 0.55 (F) |
| Abnormal CD4 count | 98.3% (58/59) | 100% (24/24) | 1.00 (F) |
| Male | 67.8% (40/59) | 33.3% (8/24) | 0.0039 * (C) |
| Female | 30.5% (18/59) | 66.7% (16/24) | 0.0024 * (C) |
| CD4 count (cells/μl) | 210[125–279] | 61 [24.5–120] | 0.0003*(MW) |
| Male | 200 [117.6–269.6] | 37.5 [7.5–181.9] | 0.0264* (MW) |
| Female | 265 [101.9–308.1] | 72.5 [29.6–141.7] | 0.0016* (MW) |
| AbnormalBMI | 45.8% (27/59) | 41.7% (10/24) | 0.73 (C) |
| Male | 33.9% (20/59) | 12.5% (3/24) | 0.0483 * (C) |
| Female | 11.9% (7/59) | 29.2% (7/24) | 0.10 (F) |
| BMI | 20 [19–21] | 18.3 [18–19] | 0.11 (MW) |
| Male | 19.8 [17.8–21] | 18.3 [17.8–20.6] | 0.80 (MW) |
| Female | 21 [19.2–23.2] | 18.4 [18–19.4] | 0.0079 * (MW) |
| Previous fractures | 8.5% (5/59) | 12.5% (3/24) | 0.68 (F) |
| Male | 6.8% (4/59) | 4.2% (1/24) | 1.00 (F) |
| Female | 1.7% (1/59) | 8.3% (2/24) | 0.20 (F) |
| Abnormal 25-Hydroxy-vitamin D | 100% (59/59) | 100% (24/24) | 1.00 (F) |
| Male | 69.5% (41/59) | 33% (8/24) | 0.0024* (C) |
| Female | 31.5% (18/59) | 66.7% (16/24) | 0.0024* (C) |
| 25-Hydroxy-vitamin D (ng/mL)[a] | 14.8±6.4 | 11.8±5.4 | 0.0466 * (T) |
| Male | 15.6±6.9 | 9.8±5.9 | 0.0324 * (T) |
| Female | 12.9±4.4 | 12.8±5.2 | 0.94 (T) |
| Abnormal FBD | 0.0% (0/59) | 0.0% (0/24) | |
| Male | 0.0% (0/59) | 0.0% (0/24) | — |
| Female | 0.0% (0/59) | 0.0% (0/24) | |

(*Continued*)

**Table 2.** (Continued)

| Parameters | ItG Mean ± SD or percentage | MiG Mean ± SD or percentage | ItG vs. MiG *p*-value |
|---|---|---|---|
| FBD | -0.2±0.8 | -0.03±0.9 | 0.37 (T) |
| Male | -0.4±0.8 | -0.2±0.5 | 0.43 (T) |
| Female | 0.3 [-0.3;0.8] | 0.1 [-0.6;0.4] | 0.31 (MW) |
| Abnormal LBD | 13.6% (8/59) | 37.5% (9/24) | 0.0324 * (F) |
| Male | 13.6% (8/59) | 20.8% (5/24) | 0.51 (F) |
| Female | 0.0% (0/59) | 16.7% (4/24) | 0.0058 * (F) |
| LBD | -0.7±1.2 | -1.2±1.4 | 0.07 (T) |
| Male | -1.0 [-1.1; -0.8] | -2.3 [-2.9; -1.3] | 0.0272 * (MW) |
| Female | 0.05±0.9 | -0.8±1.3 | 0.0291 * (T) |

* = Significant test; *T = unpaired t- test; C = chi-square test; F =* Fisher's exact test; ItG = Italian group, MiG = migrant group; Low BMD = low Bone Mineral Density; FBD = Femoral bone density evaluated with Z-score; LBD = Lumbar bone density evaluated with Z-score; BMI = Body Mass Index; MW = Mann-Whitney test; Vitamin D value used in the laboratory was 30–80 ng/mL.

greater than male Migrants (15.6 > 9.8, *p* = 0.0324). Abnormal 25-OH-vitamin D values of male Italians were most often encountered in comparison to these values of male Migrants (69.5% > 33%, *p* = 0.0024) and vice versa for female (31.5% < 66.7%, *p* = 0.0024). No significant differences between ItG and MiG groups were detected for the occurrence of a previous fracture. According to the menopause definition (amenorrhea for ≥12 consecutive months with symptoms suggestive of menopause and in which other causes of amenorrhea have been ruled out and/or the follicle-stimulating hormone (FSH) level is elevated) our female sample showed menopause status in 11/16 (68.75%) of female Migrant and 12/18 of female Italians (66%).

In **Figs 1** and **2**, the heat maps of LBD and FBD values for ItG and MiG groups are shown, respectively. Every line represents the FBD and LBD values for each patient, and the graduation of the colors are representative of the Z-score.

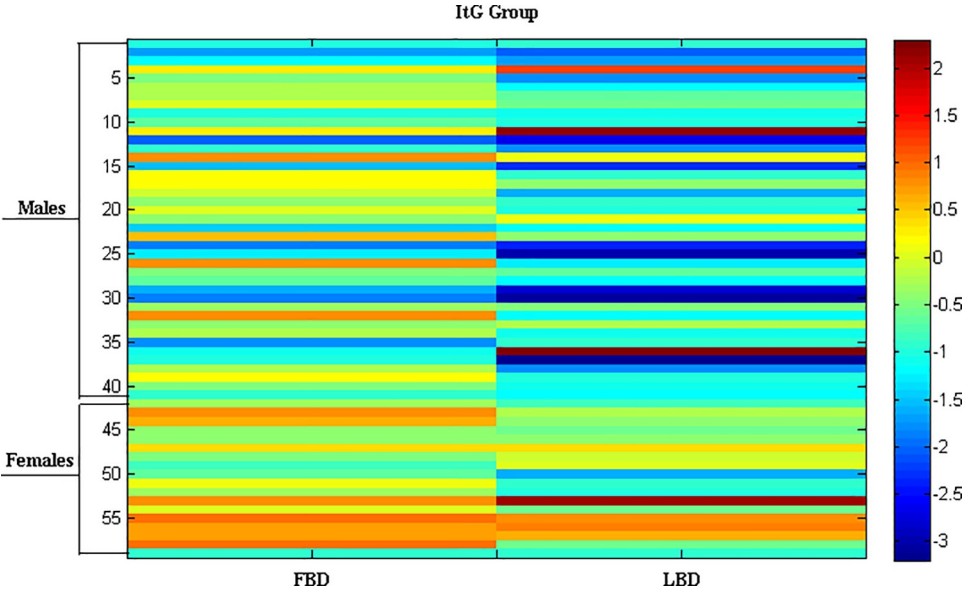

**Fig 1. Heat map of femoral BMD and lumbar BMD scores for ItG group.**

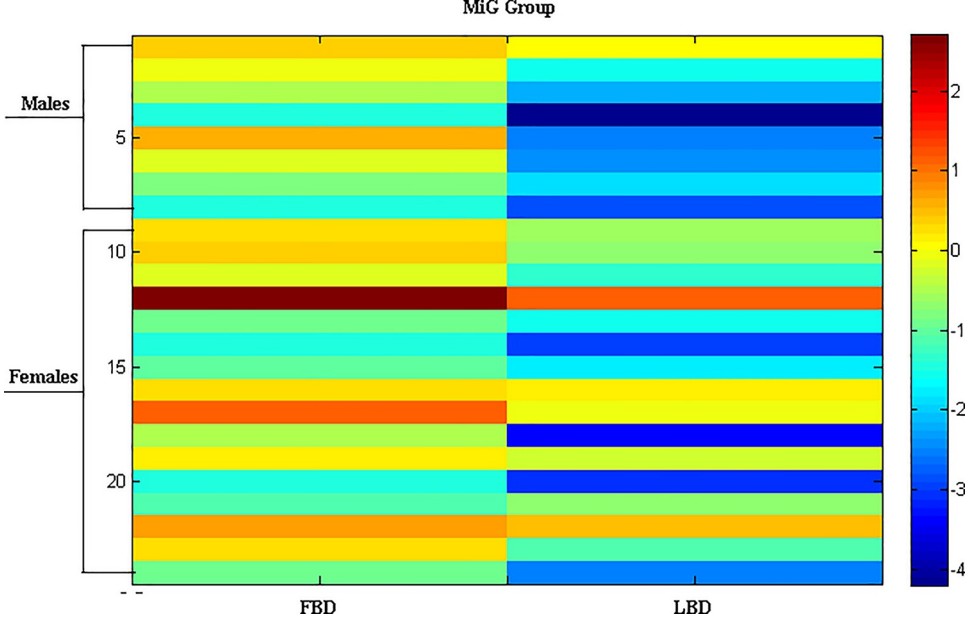

**Fig 2. Heat map of femoral BMD and lumbar BMD scores for MiG group.**

In **Fig 3**, we showed the scatter plots for ItG and MiG group between CD4 values with LBD and FBD (a-d). For every scatters plot, the red points on graphs are individuated by CD4 values and correspondent Lumbar BMD and Femoral BMD Z-scores. The blue line is the best linear fit, and the red lines define the 95% prediction interval for the regression curve. For any

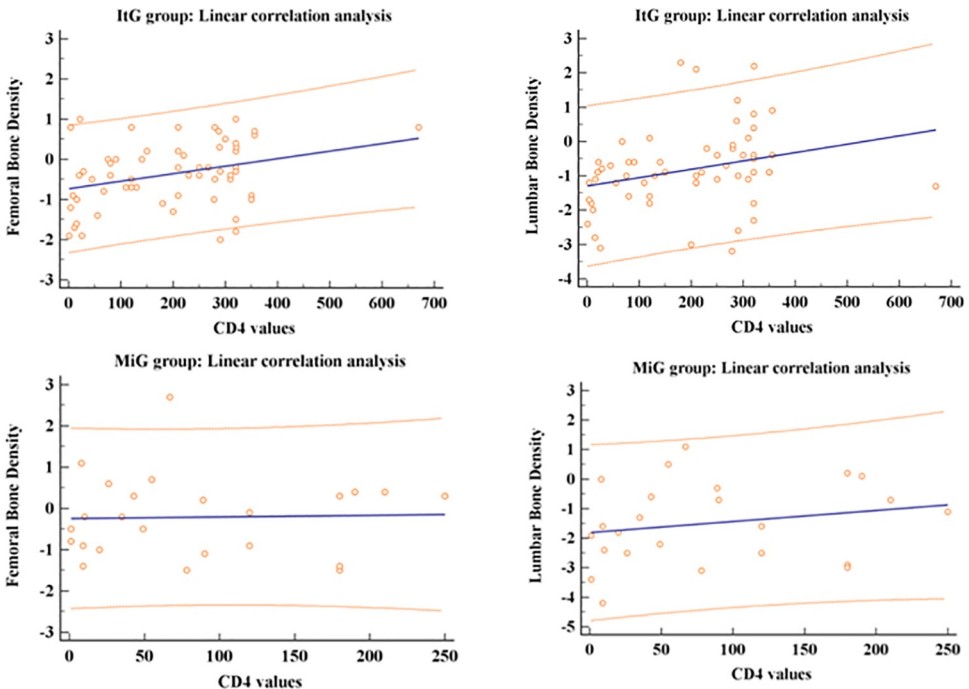

**Fig 3.** Scatter plots (a-d) for ItG and MiG group between CD4 values with Lumbar and Femoral Bone mineral densities measured with Z-score.

**Table 3. Logistic regression between *Low BMD* variable and independence variables such as gender, BMI, hydroxy-vitamin D, CD4, and previous fractures, for both ItG and MiG.**

| Logistic regression in the total sample | Coefficient | Std. Error | OR | 95% CI | p-value |
|---|---|---|---|---|---|
| Null model vs. full model | | | | | 0.0024* (C) |
| *Low BMD/* Gender | 1.3 | 0.69 | 3.8 | 0.97–14.5 | 0.056 |
| *Low BMD/* BMI | 0.07 | 0.08 | 1.07 | 0.92–1.24 | 0.39 |
| *Low BMD*/Hydroxy-vitamin D | -0.16 | 0.06 | 0.85 | 0.76–0.96 | 0.0063 * |
| *Low BMD*/CD4 | -0.004 | 0.003 | 1.0 | 0.99–1.00 | 0.17 |
| *Low BMD*/Previous fractures | 0.96 | 0.90 | 2.62 | 0.45–15.4 | 0.29 |
| Constant | -1.0 | 1.53 | — | — | 0.52 |
| **Logistic regression in ItG** | **Coefficient** | **Std. Error** | **OR** | **95% CI** | **p-value** |
| Null model vs. full model | | | | | 0.0168* (C) |
| *Low BMD/* Gender | -20.6 | 6640.7 | <0.0001 | — | 0.998 |
| *Low BMD/* BMI | 0.11 | 0.12 | 1.12 | 0.89–1.42 | 0.34 |
| *Low BMD*/Hydroxy-vitamin D | -0.169 | 0.08 | 0.84 | 0.72–0.98 | 0.0309 * |
| *Low BMD*/CD4 | -0.003 | 0.004 | 0.997 | 0.99–1.01 | 0.49 |
| *Low BMD*/Previous fractures | 0.45 | 1.33 | 1.57 | 0.12–21.3 | 0.73 |
| Constant | -0.85 | 2.09 | — | — | 0.69 |
| **Logistic regression in MiG** | **Coefficient** | **Std. Error** | **OR** | **95% CI** | **p-value** |
| Null model vs. full model | | | | | 0.13 (C) |
| *Low BMD/* Gender | 1.60 | 1.1 | 4.94 | 0.57–42.6 | 0.15 |
| *Low BMD/* BMI | -0.47 | 0–44 | 0.63 | 0.27–1.48 | 0.29 |
| *Low BMD*/Hydroxy-vitamin D | -0.13 | 0.11 | 0.88 | 0.71–1.09 | 0.23 |
| *Low BMD*/CD4 | 0.003 | 0.008 | 1.0 | 0.99–1.02 | 0.76 |
| *Low BMD*/Previous fractures | 0.97 | 1.78 | 2.63 | 0.08–86.7 | 0.59 |
| Constant | 8.76 | 7.89 | — | — | 0.27 |

* = significant test; OR = odds ratios; CI = odds ratios confidence interval at 95%; The null model = -2ln(L$_0$), where L$_0$ was the likelihood of obtaining the observations if the independent variables did not affect the outcome, the full model: -2ln(L$_0$), where L$_0$ was the likelihood of obtaining the observations with all independent variables incorporated in the model; C = chi-square test.

given value of the independent variable (CD4), this interval represents the 95% probability for the values of the dependent variable (LBD or FBD).

By linear correlation analysis, it results that for the ItG group, there was a significant positive linear correlation between CD4 and Femoral BMD ($r = 0.34$, $p = 0.009$) and between CD4 and Lumbar BMD ($r = 0.30$, $p = 0.020$). In contrast, the MiG group showed no significant correlation between CD4 and Femoral BMD ($r = 0.0$, $p = 1.0$) and between CD4 and Lumbar BMD ($r = 0.23$, $p = 0.27$). In other words, in Italian patients, an increase/decrease of CD4 values implicate an increase/decrease of Lumbar BMD or Femoral BMD scores. We did not observe these correlations in the MiG group. Finally, in Table 3, we report the logistic regression analysis between *Low BMD* variable (dichotomous) and the independent variables: Gender (dichotomous), BMI (continuous), Hydroxy-Vitamin D (continuous), CD4 (continuous), and Previous Fractures (dichotomous) for the total sample, ItG, and MiG.

For this scope, two models were considered. The null model: -2ln(L$_0$), where L$_0$ was the likelihood of obtaining the observations if the independent variables did not affect the outcome, and the full model: -2ln(L$_0$), where L$_0$ was the likelihood of obtaining the observations with all independent variables incorporated in the model. The difference between these two yields was estimated with the chi-square test to define how well the independent variables affect the outcome or dependent variable. If the chi-square test was positive ($p < 0.05$), then

there was evidence that at least one of the independent variables contributes to the prediction of the outcome.

By logistic regression, it resulted that only Hydroxy-Vitamin D was negatively correlated to *Low BMD* both in total sample (odds ratio, OR = 0.85 and $p$ = 0.0063), and in ItG (OR = 0.84 and $p$ = 0.0309). In other words, an increase (decrease) of Hydroxy-vitamin D contribute to decreasing (increasing) of *Low BMD*. In other words, a decrease of Hydroxy-vitamin D contributes to increasing the probability of osteoporosis or osteopenia in the total sample, and particularly, in the Italian patients. In the migrant group, we did not observe significant correlations by regression analysis, and the small sample size of migrants considered in our study may be the reason (see below).

## Discussion

Our study shows that "Low BMD" was significantly more frequent in the MiG group in comparison to the ItG group (migrants 37.5% vs. 13.6%). In particular, it was significantly more frequently found in migrant females than in Italian females. Our previous reports [13, 14] on the prevalence of Low-BMD in HIV mono-infected patients who underwent ARV therapy showed higher percentage rates of osteopenia (44.9%) and osteoporosis (20.9%) than an age-related healthy Italian population (18%) [16].

Our current study showed that the percentage of abnormal BMD was mostly restricted to the lumbar site. The low BMD of the lumbar site in female Migrants was more evident than in female Italians, and both males and female Migrants had a Z-score value significantly lower than males and females Italians.

Studies conducted on HIV negative patients have, indeed, shown how osteophyte formation, bone sclerosis, disk space narrowing, and spondylolisthesis are positively correlated with lumbar spine BMD. At the same time, there was no association with femoral neck BMD [31].

In general, anatomical studies suggest that sex-differences in the lumbar spine appear to be supported by postural differences in sacral bone-orientation and morphological differences in the vertebral bodies of females [32]. Moreover, emerging data show that pregnant and breast-feeding women are associated with early osteoporosis, especially in the thoracolumbar spine [12]. The involvement of the lumbar site is probably justified by the fact that our sample is younger than in other studies conducted on older populations of HIV negative subjects [13–15, 33, 34]. HCV is a risk factor for osteoporosis and fractures in the HIV population. In our studied population, HCV co-infection was found in 18.1% of the enrolled sample. Recently, one of the authors identified a more lumbar than femoral "low BMD" in HIV/HCV co-infected patients [16]. The comparison of AIDS diagnosis between Italian and migrants showed that AIDS was more frequent in Migrants, especially in females. In general, we found more frequently low CD4 values (<200 cells/µl) in both male and female Migrants in comparison with Italian patients. These data are in agreement with the emerging late diagnosis of HIV infection in young Italians comparing with age-matched female Migrants [24, 25]. By linear correlation analysis, an increase/decrease of CD4 values implicates an increase/decrease of Lumbar or Femoral BMD scores only in Italian patients. In contrast, we did not observe similar correlations in the MiG group. This aspect is probably due to the element that all Migrants had lower CD4 values than age-matched Italians. Indeed, no migrant had values of CD4 $\geq$ 350. Although the mean of BMI in Migrants was lower than Italian patients, the statistical analysis of this variable evidenced that low BMD was significant only in female Migrants.

On the other hand, the comparison of the percentage of patients with abnormal BMI was significant in male Italians. The logistic regression showed a relationship between low

Hydroxy-vitamin D and low BMD only in the Italian group. This data should be validated on a larger sample size.

Low consumption of dairy products, high consumption of soft drinks, moderate sun exposure, and a high prevalence of vitamin D deficiency have been identified among youth as risk factors for early osteoporosis. The nutrition factor is probably intrinsically associated with the geographical region and climate extremes (droughts and storms) may aggravate the bone mass loss of HIV-infected youth.

As reported in other studies, we observed early natural menopause status both in Italian than in migrant women. This finding was analogous to both groups. Apart from HIV-related immunologic status (CD4 count and viral load) other socio-demographic variables (marital status, parity, education, or income) and religious clothing of the female migrant sample may have influenced the low BMD observed in our study.

To the best of our knowledge, this study is the first DXA-based investigation to study and assess the BMD among the flow of Migrants and Refugees from countries with a high incidence of HIV infection across the Mediterranean area. In this respect, the fact that our study includes only Migrants, who were resident in Italy for no more than one year, could offer the opportunity to carefully assess the risk factors, which stress the bone composition in the HIV infected population. A recent paper on osteoporosis among HIV positive patients highlighted how an assessment of baseline bone mass loss within the first two years from the start of ARV is often lacking in longitudinal studies and claimed that there is some evidence concerning bone loss within the first year of HIV infection and ART initiation [13, 17]. The findings of a high prevalence of low BMD in HIV infected Migrants aged less than 50 years of age could suggest that the first screening for osteoporosis should be carried out as soon as possible and before the start of ART. This aspect should help the clinician in ARV management. Planning HIV therapy to prevent bone loss is crucial. We suggest the use of tenofovir alafenamide (TAF) molecule instead of tenofovir to reduce the risk of bone toxicity [35, 36]. A recent study enrolled young HIV positive with low BMD showed tenofovir as a key molecule in ARV therapy [11]. TAF is a hepatitis B virus (HBV) nucleotide reverse transcriptase inhibitor for the treatment of chronic HBV infection in adults with compensated liver disease and has been approved by the Food and Drug Administration (FDA) for the treatment of HIV-1 in November 2015.

The altered bone remodeling in young HIV positive women may suggest some alertness about our current preventive health care services. Most of the immigrants diagnosed with HIV are pregnant women who have undergone tests for sexually transmitted diseases.

Although this study is original and discloses essential steps to improve current management strategies because of new migrations from underdeveloped countries, there are some limitations. Our research involves a small size of the groups, but the cross-sectional study design still maintains its validity. This study does not permit us to establish a cause-effect relationship from our results or to evaluate the impact of osteoporosis risk factors and/or lifestyle risk factors on the management of bone disease in the HIV population. Thus, a multicentric and prospective study is warranted.

In conclusion, our study raises the question of the need to plan preventive strategies in HIV Migrants and Refugees, also in the light of the great movement of people from southern Mediterranean areas to northern European countries with low sun exposure [21, 22]. The female Migrants must be protected from precocious low BMD with an additional layer of safety [17–19, 21, 22, 33]. The approach requires different and specific intervention strategies in the future. Our data confirm that early screening for low BMD and other risk factors associated with bone loss in HIV patients is useful [2, 7, 15, 18–20].

In agreement with the osteoporosis foundation (http://share.iofbonehealth.org/WOD/Compendium/2019-IOF-Compendium-of-Osteoporosis-PRESS.pdf) the authors stress the strategies for preventing osteoporosis. The role of nutrition in maintaining bone health associated with a personalized anti-HIV treatment and daily vitamin supplementation to reduce early bone loss as well as a baseline DXA are critical steps for HIV Migrants and Refugees after arrival in the host countries [13,17,20,23,36]. High-risk groups should be first identified, and osteoporosis investigations should be carried out earlier than at 50 years of age [37–39].

## Limitations

As indicated above, our results should be evaluated with caution due to the retrospective nature of the study. Also, despite the estimated minimum sample sizes of ItG and MiG were correctly applied and the conditions for the sample size according to Bernoulli are met, prospective and more extensive studies are needed to confirm our results.

## Acknowledgments

The authors AC, PDC, and NS, contributed equally to this paper. We are deeply grateful to the patients and their families for participating in the study.

## Disclosure

Preliminary results of this study were presented at the HIV Glasgow Conference, Oct. 23–26, 2016, Glasgow, United Kingdom.

## Author Contributions

**Conceptualization:** Antonio Cascio, Paola Di Carlo.

**Data curation:** Antonio Cascio, Paola Di Carlo, Nicola Serra, Giuseppe Lo Re, Antonio Lo Casto, Consolato Sergi.

**Formal analysis:** Antonio Cascio, Claudia Colomba, Paola Di Carlo, Giuseppe Lo Re, Angelo Gambino, Giuseppe Guglielmi, Nicola Veronese, Roberto Lagalla.

**Funding acquisition:** Paola Di Carlo.

**Investigation:** Antonio Cascio, Giuseppe Guglielmi, Consolato Sergi.

**Methodology:** Claudia Colomba, Paola Di Carlo, Giuseppe Lo Re, Angelo Gambino, Antonio Lo Casto, Roberto Lagalla.

**Project administration:** Antonio Cascio, Claudia Colomba, Nicola Veronese, Roberto Lagalla.

**Resources:** Giuseppe Lo Re, Angelo Gambino, Antonio Lo Casto, Giuseppe Guglielmi.

**Software:** Nicola Serra, Nicola Veronese, Roberto Lagalla.

**Supervision:** Angelo Gambino.

**Validation:** Antonio Cascio, Paola Di Carlo, Nicola Serra, Antonio Lo Casto, Nicola Veronese, Consolato Sergi.

**Visualization:** Giuseppe Guglielmi.

**Writing – original draft:** Antonio Cascio, Claudia Colomba, Paola Di Carlo.

**Writing – review & editing:** Paola Di Carlo, Nicola Serra, Giuseppe Lo Re, Angelo Gambino, Antonio Lo Casto, Giuseppe Guglielmi, Nicola Veronese, Roberto Lagalla, Consolato Sergi.

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
