## [Decision Letter · Decision Letter 0]

22 Apr 2020

PONE-D-20-08720

Osteoporosis in HIV-positive Young Italians and Migrants

PLOS ONE

Dear Prof. Sergi,

Thank you for submitting your manuscript to PLOS ONE. After careful consideration, we feel that it has merit but does not fully meet PLOS ONE’s publication criteria as it currently stands. Therefore, we invite you to submit a revised version of the manuscript that addresses the points raised during the review process.

As you can find from the comments noted below, the authors should address samples size and provide details of the study like exclusion and inclusion criteria of subjects with a clear hypothesis.  The second reviewer also suggested the manuscript should be proof read for English language.   

We would appreciate receiving your revised manuscript by Jun 06 2020 11:59PM. To enhance the reproducibility of your results, we recommend that if applicable you deposit your laboratory protocols in protocols.io, where a protocol can be assigned its own identifier (DOI) such that it can be cited independently in the future. For instructions see: http://journals.plos.org/plosone/s/submission-guidelines#loc-laboratory-protocols

We look forward to receiving your revised manuscript.

Kind regards,

Dr. Sakamuri V. Reddy

Academic Editor

PLOS ONE

Journal Requirements:

'This study was funded by an intramural grant of the University of Palermo, Italy.'

'The authors received no specific funding for this work.'

Reviewers' comments:

Reviewer's Responses to Questions

**Comments to the Author**

1. Is the manuscript technically sound, and do the data support the conclusions?

Reviewer #1: Yes

Reviewer #2: No

2. Has the statistical analysis been performed appropriately and rigorously? 

Reviewer #1: No

Reviewer #2: No

3. Have the authors made all data underlying the findings in their manuscript fully available?

Reviewer #1: Yes

Reviewer #2: No

4. Is the manuscript presented in an intelligible fashion and written in standard English?

Reviewer #1: Yes

Reviewer #2: No

5. Review Comments to the Author

Reviewer #1: Thank you for inviting me to review this manuscript which compares the prevalence of OP in HIV positive italians versus migrants.

Overall, the manuscript is well written. Below are a few suggestions that may improve the manuscript.

Introduction

The paragraph regarding OP, its definition and associated factors is unnecessary. The OP should relate the prevalence of OP to HIV + individuals instead. The authors should also perform a review of whether previous studies have looked into the prevalence of OP in HIV+ pts in Italy prior to this.

Methods

Sample size is rather small. Was sample size calculation performed? was data normally distributed? If data not normally distributed, continuous variables should be presented as median and IQR (refer table 1). Please state the median and IQR for BMI as well. Non parametric tests should be used for the analyses as well.

BMD assessment

For younger age pts, it has been recommended that the z-score is used instead of the t-score.

Results

Table 1 and 2 can be merged. Results can be summarised more succintly. Highlight the importance of the results, rather than repeat what is already presented in the table.

Why was logistic regression not performed? It would be interesting to see the OR of a italian or migrant developing OP.

Discussion.

The first para of the discussion should highlight the most important findings, followed by discussion of the results. A lot of the material (regarding past similar studies done in Italy) in the first para should be moved to introduction. The discussion should also be shortened. It's currently a very long read.

Limitations should be added regarding the small sample size

Reviewer #2: General Comments:Much better suited (after major revision) for an HIV-specific journal – as there is little to be learned for a general medical audience from this study. Would rewrite this with the help of a native English speaker into a much more descriptive paper – not “hunting” for statistical significance.

Specific Comments:

(Relatively) small comparative cross-sectional study on bone mineral density in HIV-infected, ART-naïve (mostly African) recent migrants and native Italians from Sicily.

The major problems with this study are:

- no underlying hypothesis

- no clear inclusion/exclusion criteria (who got a DXA scan and why?) what was the exact age range in each group?

- the major “finding” was a much higher rate of osteoporosis/osteopenia in migrant women compared with Italian women – that seemed to be explained by low rate of osteopenia in Italian women in their sample and likely also by much more advanced HIV disease in migrant women – but the authors unfortunately did not delineate HIV-specific and other osteoporosis by gender and migrant status. Important questions: menopause status? History of TB?

Minor points: see comments in edited .pdf file.

- Would elaborate on implications for ART management and longitudinal monitoring in discussion and discuss osteoporosis treatment

6. PLOS authors have the option to publish the peer review history of their article (what does this mean?). If published, this will include your full peer review and any attached files.

Reviewer #1: No

Reviewer #2: No

---

## [Author Response · Author response to Decision Letter 0]

10 Jun 2020

'This study was funded by an intramural grant of the University of Palermo, Italy.'

Dear Editor, we did not provide any information in section acknowledgment because The authors received intramural funding (University of Palermo) for this work.

'The authors received no specific funding for this work.'

a. Please clarify the sources of funding (financial or material support) for your study. List the grants or organizations that supported your study, including funding received from your institution.

b. State what role the funders took in the study. If the funders had no role in your study, please state: "The funders had no role in study design, data collection, and analysis, decision to publish, or preparation of the manuscript."

d. If you did not receive any funding for this study, please state: "The authors received no specific funding for this work."

 

Reviewer #1: Thank you for inviting me to review this manuscript, which compares the Prevalence of O.P. in HIV positive Italians versus migrants.

Overall, the manuscript is well written. Below are a few suggestions that may improve the manuscript.

Reviewer #1 comment 

• Introduction

The paragraph regarding O.P., its definition, and associated factors are unnecessary. The O.P. should relate the Prevalence of O.P. to HIV + individuals instead. The authors should also perform a review of whether previous studies have looked into the Prevalence of O.P. in HIV+ pts in Italy prior to this.

Response 

Thank you very much for your comments and suggestions that helped us to improve the manuscript.

The paragraph regarding O.P. remains in the introduction according to the comments of reviewer 2. However, the authors have been introduced the Prevalence of O.P. in HIV individual in Italy via inserted the following sentences and reference number 16.

Several studies suggest that fracture rates are higher in populations with HIV than among matched uninfected controls despite ART and gender (9-11). The causes of a low BMD could be different in HIV‐infected patients by considering the time of HIV infection �12�.Previous studies conducted in our geographical area founded that the Prevalence of osteoporosis was significantly higher in HIV- infected than in uninfected subjects, a result similar to previous meta-analyses [12-15]. The Prevalence of osteopenia and osteoporosis in HIV mono-infected patients in our geographical area was about 44.9%, and 20.9% in comparison with 18% reported in a healthy Italian population �16

Cavalli L, Guazzini A, Cianferotti L, et al. Prevalence of osteoporosis in the Italian population and main risk factors: results of BoneTour Campaign. BMC Musculoskelet Disord. 2016;17(1):396. Published 2016 Sep 17. doi:10.1186/s12891-016-1248-8

Reviewer #1 comment 

Methods

The sample size is rather small. Was sample size calculation performed? was data normally distributed? If data not normally distributed, continuous variables should be presented as median and IQR (refer table 1). Please state the median and IQR for BMI as well. Non parametric tests should be used for the analyses as well. 

According to the suggestions of the reviewer the authors we have thoroughly modified the following sections 

Materials and Methods

The Mann-Whitney test was used to test the difference between two independent samples (ItG and MiG group). It was the alternative for the independent samples t-test, when the distribution of the samples is not Normal, Tests for Normal distribution were performed with the D'Agostino-Pearson test.

Sample size estimation

To individualize a sample size statistically significant in this study, we considered a Bernoulli sampling (Strand, M. M.,1979) for both Italian and Migrant groups.

Strand, M. M., Estimation of a population total under a "Bernoulli sampling" procedure. The American Statistician, 33(2), 81-84, (1979).

For ItG, the minimum sample size for this study was estimated equal to 41 patients affected by low bone mass density. It was obtained considering a statistical z-score at 95%, an error � = 15% and hypothesizing a prevalence �, about 60% according to the studies of Cavalli L. et al. (2016) and Tomažič, J., et al. (2007), in this way we estimated a prevalence range equal to 45%-75% 

Cavalli L., Guazzino A., Cianferotti L., Parri S., Cavalli T., Metozzi A., Giusti F., Fossi C., Black DM, and Brandi ML., Prevalence of osteoporosis in the Italian population and main risk factors: results of BoneTour Campaign. BMC Muscoloskeletal Disorders, 2016, 17:396. DOI 10.1186/s12891-016-1248-8

Tomažič, J., Ul, K., Volčanšek, G., Gorenšek, S., Pfeifer, M., Karner, P., ... & Vidmar, L. (2007). Prevalence and risk factors for osteopenia/osteoporosis in an HIV-infected male population. Wiener Klinische Wochenschrift, 119(21-22), 639-646.

For MiG, the minimum sample size for this study was estimated equal to 22 patients affected by low bone mass density. It was obtained considering a z-score at 95%, an error � = 20% and hypothesizing a prevalence � about 35%, according to Varenna, M. et al. (2003), in this way we estimated a prevalence range equal to 15%-55%. In this case, we considered an error � more significant to an error in ItG because for migrants in Italy there were harboring less information about bone mass density disease 

Varenna, M., Binelli, L., Zucchi, F., Rossi, V., & Sinigaglia, L. (2003). Prevalence of osteoporosis and fractures in a migrant population from southern to northern Italy: a cross-sectional, comparative study. Osteoporosis international, 14(9), 734-740.

Also, the sample size for ItG and MiG was enlarged to 59 and 24 patients, respectively, considering the possibility of unexpected events and, consequently, the possibility of patients data loss.

Statistical analysis

The statistical analysis was performed by MATLAB statistical toolbox version 2008 (MathWorks, Natick, MA, USA). Data are presented as numbers and percentages for categorical variables. Numerical data are expressed as the mean ± standard deviation (S.D.) or median and confidence interval at 95% (CI). The �2 test and Fisher's exact tests were performed to evaluate significant differences of proportions or percentages between two groups. Fisher's accurate test was used where the �2 test was not appropriate. The Mann-Whitney test was used to test the difference between two independent samples (ItG and MiG group). It was the alternative for the independent samples t-test when the distribution of the samples is not Normal, Tests for Normal distribution were performed with the D'Agostino-Pearson test.

Linear correlation analysis was performed, and the test on Pearson's linear correlation coefficient r was performed with the t-Student test, under the null hypothesis of Pearson's linear correlation coefficient r = 0. 

The logistic regression was performed to analyze the relationship between Low Bone Density (dichotomous variable) and the independent variables: Gender (dichotomous – M=1, F=0), Previous fractures (dichotomous – yes=1, no =0), BMI (continuous), CD4 cells (continuous), and 25-Hydroxy-vitamin D (continuous). Finally, all tests with p-value < 0.05 were considered significant. 

Reviewer comment 

BMD assessment

For younger age pts, it has been recommended that the z-score is used instead of the t-score.

Thank you very much for this observation. According to the suggestion of the reviewer, the authors have calculated the Z scores to evaluate BMD and added the following sentences 

For each scan, BMD, Z-scores were recorded as previously reported [14, 16, 17]. DXA measurements were performed in the femur (femoral neck and/or total hip) and lumbar spine in each patient. Since the age of our patients ranged from 30 to 50 years, the use of Z-scores (defined as an individuals' BMD in comparison to age-matched normal individuals) was used for all the analyses, according to World Health Organisation (WHO) recommendation. Participants were categorized as having low BMD if the femoral neck or total lumbar spine Z-score was– 2 or less

 

Results

Reviewer comment 

Tables 1 and 2 can be merged. 

Response 

Thank you for your suggestion. We tried to merge, but we found that table would become populated with a lot of data. Large tables are difficult to follow with the text. Thus, we would kindly prefer to leave the two tables separated. Moreover, we also entered new statistical tests as suggested by the reviewer, and this would make it difficult to arrange the two tables in one.

Results can be summarised more succinctly. Highlight the importance of the results, rather than repeat what is already presented in the table.

Why was logistic regression not performed? It would be interesting to see the OR of an Italian or migrant developing O.P.

Thank you very much for this crucial suggestion. As suggested by the reviewer, the logistic regression is showed in table 3 and the results thoroughly discussed 

Please see the statistical section 

The logistic regression was performed to analyze the relationship between Low Bone Density (dichotomous variable) and the independent variables: Gender (dichotomous – M=1, F=0), Previous fractures (dichotomous – yes=1, no =0), BMI (continuous), CD4 cells (continuous), and 25-Hydroxy-vitamin D (continuous). Finally, all tests with p-value < 0.05 were considered significant.

Results section:

Table 3. Logistic regression between Low BMD variable and independence variables such as Gender, BMI, Hydroxy-vitamin D, CD4, and Previous fractures, for both ItG and MiG

Logistic regression in total sample Coefficient Std. Error OR 95% CI p-value

Null model vs. full model 0.0024* (C)

Low BMD/ Gender 1.3 0.69 3.8 0.97-14.5 0.056

Low BMD/ BMI 0.07 0.08 1.07 0.92-1.24 0.39

Low BMD/Hydroxy-vitamin D -0.16 0.06 0.85 0.76-0.96 0.0063 *

Low BMD/CD4 -0.004 0.003 1.0 0.99-1.00 0.17

Low BMD/Previous fractures 0.96 0.90 2.62 0.45-15.4 0.29

Constant -1.0 1.53 � � 0.52

Logisticregression in ItG Coefficient Std. Error OR 95% CI p-value

Null model vs. full model 0.0168* (C)

Low BMD/ Gender -20.6 6640.7 <0.0001 � 0.998

Low BMD/ BMI 0.11 0.12 1.12 0.89-1.42 0.34

Low BMD/Hydroxy-vitamin D -0.169 0.08 0.84 0.72-0.98 0.0309 *

Low BMD/CD4 -0.003 0.004 0.997 0.99-1.01 0.49

Low BMD/Previous fractures 0.45 1.33 1.57 0.12-21.3 0.73

Constant -0.85 2.09 � � 0.69

Logistic regression in MiG Coefficient Std. Error OR 95% CI p-value

Null model vs. full model 0.13 (C)

Low BMD/ Gender 1.60 1.1 4.94 0.57-42.6 0.15

Low BMD/ BMI -0.47 0-44 0.63 0.27-1.48 0.29

Low BMD/Hydroxy-vitamin D -0.13 0.11 0.88 0.71-1.09 0.23

Low BMD/CD4 0.003 0.008 1.0 0.99-1.02 0.76

Low BMD/Previous fractures 0.97 1.78 2.63 0.08-86.7 0.59

Constant 8.76 7.89 � � 0.27

* = significant test; OR = odds ratios; CI = odds ratios confidence interval at 95%; The null model= -2ln(L0), where L0 was the likelihood of obtaining the observations if the independent variables did not affect the outcome, the full model: -2ln(L0), where L0 was the likelihood of obtaining the observations with all independent variables incorporated in the model; C = chi-square test

For this scope, two models were considered. The null model: -2ln(L0), where L0 was the likelihood of obtaining the observations if the independent variables did not affect the outcome, and the full model: -2ln(L0), where L0 was the likelihood of obtaining the observations with all independent variables incorporated in the model. The difference between these two yields was estimated with the chi-square test, to define how well the independent variables affect the outcome or dependent variable if chi-square test was positive (p-value < 0.05), then there was evidence that at least one of the independent variables contributes to the prediction of the outcome.

By logistic regression, it resulted that only Hydroxy-vitamin Dwas negatively correlated to Low BMD both in the total sample (OR = 0.85 and p-value = 0.0063), and in ItG(OR = 0.84 and p-value = 0.0309), in other words, an increase(decrease) of Hydroxy-vitamin D contribute to decrease(increase) of Low BMD, i.e., according to other studies, a decrease of Hydroxy-vitamin D, contribute to increase the probability of osteoporosis or osteopenia in the total sample, and in particular, in Italian patients. In addition, we performed the regression analysis in the migrant group, but no significant correlations were found. This could be due to the small sample size of migrant considered in our study

And the discussion section 

The logistic regression showed a relationship between low Hydroxy-vitamin D and low bone mineral density only in the Italian group, while in migrants this analysis was not significant 

Therefore, this result should be verified on more large sample size. 

 

• Reviewer comment 

Discussion.

The first paragraph of the discussion should highlight the most important findings, followed by discussion of the results. A lot of the material (regarding past similar studies done in Italy) in the first para should be moved to introduction. The discussion should also be shortened. It's currently a very long read.

Limitations should be added regarding the small sample size

Response 

Thank you for this stylistic suggestion. As suggested by the reviewer, the discussion section was revised, and the limitation section introduced 

Reviewer #2: General Comments: 

Much better suited (after major revision) for an HIV-specific journal – as there is little to be learned for a general medical audience from this study. Would rewrite this with the help of a native English speaker into a much more descriptive paper – not "hunting" for statistical significance.

Specific Comments:

(Relatively) small comparative cross-sectional study on bone mineral density in HIV-infected, ART-naïve (mostly African) recent migrants and native Italians from Sicily.

The major problems with this study are:

- no underlying hypothesis

- no clear inclusion/exclusion criteria (who got a DXA scan and why?) what was the exact age range in each group?

- the major "finding" was a much higher rate of osteoporosis/osteopenia in migrant women compared with Italian women – that seemed to be explained by low rate of osteopenia in Italian women in their sample and likely also by much more advanced HIV disease in migrant women – but the authors unfortunately did not delineate HIV-specific and other osteoporosis by gender and migrant status. Important questions: menopause status? History of T.B.?

Minor points: see comments in edited .pdf file.

- Would elaborate on implications for ART management and longitudinal monitoring in discussion and discuss osteoporosis treatment

The major problems with this study are:

- no underlying hypothesis

Response

Thank you for the suggestions in improving this manuscript. The studies on low bone mineral density in the young healthy immigrant population are very scattered. They have been often performed in non-specific age-matched communities, which cannot give the statistic results a strong power. They are often made by a small simple size. Each study includes a breed with different genetic characteristics as reported below by the authors; the female gender has also been studied in the phase preceding menopause. It is clear that with these thin reference data on the Prevalence of female immigrant population, HIV positive is difficult, but we reformulate the hypothesis and the introduction. 

Several studies suggest that fracture rates are higher in populations with HIV than among matched uninfected controls despite ART and gender (9-11). The causes of a low BMD could be different in HIV‐infected patients by considering the time of HIV infection �12�.Previous studies conducted in our geographical area found that the Prevalence of osteoporosis was significantly higher in HIV- infected than in uninfected subjects, a result similar to previous meta-analyses [12-15]. The Prevalence of osteopenia and osteoporosis in HIV mono-infected patients in our geographical area was about 44.9%, and 20.9% in comparison with 18% reported in a healthy Italian population�16

A recent analysis of bone-healthy in immigrant populations conducted in Sweden showed that women had low BMD for age according to the American and African – American referents (17-19). Moreover, a study conducted in German-Turkish immigrants osteopenia was diagnosed in 32% and osteoporosis in 8% of young migrants. (20). Lifestyle, BMI, and vitamin D levels may be influenced by low BMD in the migrant setting. Recently low level of vitamin D were reported in Somali migrant women in Sweden and migrants in Canada (21-23)

Although the number of migrants and refugees crossing the Mediterranean Sea has decreased in 2018 and 2019, this number has been unprecedented in the last ten years compared with the past twenty or thirty years [21,22]. Recent Italian data showed that the new diagnosis of HIV infection in 72% of migrants was late and less than six months before developing AIDS 

Half came from Sub-Saharan Africa (SSA, 59.4%) and showed an increasing percentage of fertile female �22,23�.

The incidence of bone mineral density loss and related fractures attributed to a specific eating pattern has also been targeted for guidelines and suggestions within the EU/EEA [21,22,25]. 

Recently, Chisati et al. �11� reported an early loss of BMD in HIV infected patients treated with tenofovir also in young subjects; according to with this data and take in account the emerging of late diagnosis of HIV Italian migrants �24�the authors have studied the burden of osteoporosis in naïve ARV HIV positive patients and compare the bone density of the native Italian population group (ItG) with that of HIV Migrants (MiG) upon arrival in Italy.

Editor's comment

- no clear inclusion/exclusion criteria (who got a DXA scan and why?) 

Response: 

Thank you for this crucial aspect. Dual-energy X-ray absorptiometry (DEXA) is globally accepted as a standard technique for measuring BMD performed typically at the lumbar spine and femoral neck [1,2]. 

1. World Health Organization. Assessment of fracture risk and its application to screening for postmenopausal osteoporosis. WHO technical report series 843. Geneva: WHO, 1994.

Kanis JA, McCloskey EV, Johansson H, Oden A, Melton LJ, Khaltaev N. A reference standard for the description of osteoporosis. Bone 2008; 42:467–475. 

Moreover, according to the suggestion of the reviewer number 1 the authors have been revised the manuscript and participants were categorized as having low BMD if the femoral neck or total lumbar spine Z-score was– 2 or less as reported by other studies as the following 

1. Chisati EM, Constantinou D, Lampiao F (2020) Reduced bone mineral density among HIV infected patients on antiretroviral therapy in Blantyre, Malawi: Prevalence and associated factors. PLOS ONE 15(1): e0227893. https://doi.org/10.1371/journal.pone.0227893

Editor's comment 

What was the exact age range in each group?

Response

Thank you for the critical data that was missing. In table 1 and the materials and methods section the authors introduced the age of the enrolled population 

The ItG (Italian group), composed of 59 patients, with ages into range 30-49, mean

43.7 years old and standard deviation (S.D.) equal to 4.8 years old.

.

The MiG (migrant group), composed of 24 patients, with ages into range 32-50, mean

45.4 years old, and standard deviation (S.D.) equal to 4.8 years old. In detail, 22/24

(92%) migrants came from Africa and 2/24 (8%) from Asia.

 

Minor revision request 

Highlights section 

sentence: naïve antiretroviral treatment

Reviewer 2 comment

• word order

Response : 

Thank you! We changed into HIV positive patients naive to antiretroviral therapy

reviewer 2 comment

would rephrase HIV positive migrants are younger than 50 years and, especially if female, are at risk of osteoporosis 

Response: Thank you! The authors, as suggested by the reviewer, have modified the sentence as follows "even if younger than 50 years are at risk for osteoporosis, especially if they are female."

Introduction section 

Reviewer 2 comment

• Consider omitting. Well known.

According to the reviewer's suggestion, the authors have deleted the following sentence 

<<Antiretroviral treatment (ARV) has impacted positively on morbidity and mortality among

human immunodeficiency virus (HIV)-positive patients.>>

Reviewer 2 comment

• The demographic and cultural heterogeneity of the HIV-infected population in terms of race and lifestyles show how complex is the debate about the prevention and management of the bone disease 

• Would try to rephrase that with the help of a native English speaker

According to the suggestion of the reviewer, the authors have modified the structure of the introduction section and inserted the following sentences. 

<< Recent analysis on bone healthy in immigrant population conducted in Sweden showed that women had low BMD for age according to the American and African – American referents (17-19). Moreover, a study conducted in German-Turkish immigrants osteopenia was diagnosed in 32% and osteoporosis in 8% of young migrants. (20). Low BMD may influence lifestyle, BMI, and vitamin D level in migrant setting. Recently, low level of vitamin D was reported in Somali migrant women in Sweden and migrants in Canada (21-23)>>

We added some crucial references also from the same Province where the senior author is resident.

We also added the following references 

17. Demeke T, Osmancevic A, Gillstedt M, et al. Comorbidity and health-related quality of life in Somali women living in Sweden. Scand J Prim Health Care. 2019;37(2):174–181. doi:10.1080/02813432.2019.1608043

18. Mgodi, N.M., Kelly, C., Gati, B. et al. Factors associated with bone mineral density in healthy African women. ArchOsteoporos 10, 3 (2015). https://doi.org/10.1007/s11657-015-0206-7

19. Chantler, S., Dickie, K., Goedecke, J.H. et al. Site-specific differences in bone mineral density in black and white premenopausal South African women. Osteoporos Int 23, 533–542 (2012).

20. Klemm, P., Dischereit, G. & Lange, U. Adult lactose intolerance, calcium intake, bone metabolism and bone density in German-Turkish immigrants. J Bone MinerMetab 38, 378–384 (2020). https://doi.org/10.1007/s00774-019-01070-https://doi.org/10.1007/s00198-011-1570-9

21. https://www.migrationdataportal.com/data?i=stock_abs_&t=2019

22. https://www.ecdc.europa.eu/sites/default/files/documents/HIV-migrants-Monitoring

23. Vitamin D status of refugees arriving in Canada. Michael Aucoin, Rob Weaver, Roger Thomas, Lanice Jones. Canadian Family Physician Apr 2013, 59 (4) e188-e194;

• A real tragedy but not really related to the topic....

According to the suggestion of the reviewer, the authors have deleted the following sentence:

<< By March 2016, the number of refugees and migrants who had reached Europe by sea was 171,000, missing or had lost their lives in the process (UNHCR) [7].and 711 were missing or had lost their lives in the process (UNHCR) [7]>>

• Try to shorten this section, focusing on how late presentation, low CD4, malnutrition, low Vit D impact bone density. Remember, this is not an essay about the refugee crisis...

Yes, indeed. You are right and, even we are facing a humanitarian crisis, it is not part of the topic, and we deleted this part. According to the suggestion of the reviewer, the authors have modified the sentences as follows: 

Recent Italian data showed that the new diagnosis of HIV infection in 72% of migrants was late and less than six months before developing AIDS Half came from Sub-Saharan Africa (SSA, 59.4%) and showed an increasing percentage of fertile female �22-25�.

Reviewer 2 comment

• too wordy. Would focus on how osteoporosis impacts and is impacted by general health, mortality, etc...

this field was present in the top of introduction and in the new sentences about migrant healthy status please see as reported in the previously sentence about HIV negative and HIV positive bone healthy and vitamin D deficiency 

• Would move this more to the top of the introduction

Thank you for your beneficial suggestion. According to the advice of the reviewer, the introduction was re-arrenged.

• Nice fact about Sicily but not really relevant in this context

Response 

According to the suggestion of the reviewer, the corresponding sentences were deleted.

• would elaborate more if you feel that this is relevant for osteoporosis among younger people in Sicily.

According to the suggestion of the reviewer, the sentences were deleted because as was showed in the central part of the introduction this is relevant for all naïve HIV positive patients especially migrants who confirmed HIV late diagnosis 

• Did you have a hypothesis when you undertook this study? If so would state so and move it to the center of the introduction...

According to the suggestion of the reviewer, the authors have shown in the center of the introduction as the low BMD is observed in migrants HIV negative and the changing trend of Italian migrants characterized by new late diagnosis of HIV and a higher presence of fertile women 

 

Materials and methods 

• why did they have this test?

• Was this a fallout from another prospective study?

• Was DXA common practice at your clinic, even in below 50 year olds?

-- Need to explain as there is selection bias looming....

The authors aimed to verify if the low bone mineral density is precocious and related to AIDS condition and relevant to the status of migrants. All relevant references were cited. Osteopenia and osteoporosis seem to be more frequent in young HIV negative migrants. DXA is used in our clinic and was adopted for this study following approval of the Ethics Committee of the University of Palermo.

• What were the exact inclusion/exclusion criteria? Did they have to be <50 years of age? Were there BMI criteria? Anybody with wasting syndrome or on steroids?

Thank you for this aspect, and we clarified better the inclusion/exclusion criteria in the manuscript. We retrospectively analyzed all patients who underwent to DEXA analysis within six months from HIV diagnosis independent of their BMI; we excluded all pts with the treatment of steroid‐induced bone loss and active tuberculosis 

• too specific

25-Hydroxy-vitamin D was assayed as previously reported 

We added the reference of Li Vecchi V, Soresi M, Giannitrapani L, et al. Dairy calcium intake and lifestyle risk factors for bone loss in HIV-infected and uninfected Mediterranean subjects. BMC Infect Dis. 2012;12:192. Published 2012 Aug 15. doi:10.1186/1471-2334-12-192

• This sentence typically goes to the Results section

Thank you for your suggestion. This sentence was included in the result section 

• same here

The sentence was also included in the result section 

• no need to spell out these definitions...

As suggested by the reviewer, the authors have deleted this sentence

• remove. There are no animal data in this study.

According to the suggestion of the reviewer, the authors have deleted the sentences and insert the following sentences at the top of the results section

<<The 83 patients, composed by 59.4% males and 40.6% females, with ages into range 30-50, mean 44.2 years old and standard deviation (S.D.) equal to 4.9 years old, was subdivided into two groups: The ItG (Italian group), composed of 59 patients, with ages into range 30-49, mean 43.7 years old and standard deviation (S.D.) equal to 4.8 years old. The MiG (migrant group), composed of 24 patients, with ages into range 32-50, mean 45.4 years old, and standard deviation (S.D.) equal to 4.8 years old. In detail, 22/24 (92%) migrants came from Africa and 2/24 (8%) from Asia.>>

• did you statistically correct for multiplicity of analyses?

As suggested by reviewer number 1, all statistical analysis have been revised, and new statistical test were introduced 

• this abbreviation has not been explained yet. I assume it's lumbar spine...

Yes, indeed. According to the new analysis of low bone mineral density, participants were categorized as having low BMD if the femoral neck or total lumbar spine Z-score was– 2 or less 

the abbreviation was Low bone mineral density 

• same as for "LBD"

Thank you, we changed according to the reviewer's suggestion.

• this is the most important finding. But is it just explained by more advanced HIV?

We interpret your comment, and we separated the patients by gender.

• Did migrant women have more advanced HIV ? 

By linear correlation analysis, it results that for ItG group, there was a significant positive linear correlation between CD4 and Femoral BMD (r =0.34,p-value = 0.009) and between CD4 and Lumbar BMD (r = 0.30, p-value = 0.020), instead for MiG group there was not a significant correlation between CD4 and Femoral BMD (r =0.0,p-value = 1.0) and between CD4 and Lumbar BMD (r = 0.23, p-value = 0.27). In other words, in Italian patients, an increase/decrease of CD4 values implicate an increase/decrease of Lumbar BMD or Femoral BMD scores. In contrast, we did not observe these correlations in the MiG group.

• what about if stratified by gender and migrant status?

In the top of the result section we specified that we found major presence of migrant females with abnormal lumbar value in comparison to Italian females (16.7% > 0.0%, p= 0.0058).

• because the n was smaller, the difference was even bigger...would consider leaving away

• Table 1 

As suggested by the reviewer the authors have modified the table 1 and table 2.

Parameters % (Number)

Patients 83

Italian

 male

 female 71.1% (59)

49.4% (41)

21.7% (18)

Migrant

 male

 female 28.9% (24)

9.6% (8)

19.3% (16)

Age, in years 

 mean ±SD

 Range 

44.2 ±4.9 

30-50

Abnormal BMI 49.4% (41)

CD4 count ≤ 500 (cells/μl) 98.8% (82)

CD4 count (cells/μl)

 <200

 200-349

 ≥350 

55.4% (46)

41.0% (34)

3.6% (3)

Low level of 25-Hydroxy-vitamin D 100% (83)

AIDS 56.6% (47)

Mono-infected (HIV) 81.9% (68)

Co-infected (HIV and HCV) 18.1% (15)

Low BMD (ItG+MiG) 20.1% (17=8+9)

Previous fractures 9.6% (8)

BMI= body mass index: normal range [18,5-24,9]; 

CD4 count: normal range [500-1500] (cells/μl); 

Vitamin D: normal range [30-100] (ng/mL)a; 

BMD = Bone Mineral Density

Tabella 2 

According to the suggestion of the reviewer number 1 all the table 2 was modified as follows: 

Parameters ItG

Mean ± S.D. or percentage MiG

Mean ± S.D. or percentage 

ItG vs. MiG

p-value

Nr. Patients 59 24 

Age

 Male

 Female 46 [43-47]

44[41-47]

47[44.8-47] 47.5[43-49]

44.5[37.8-49.2]

47.5[44.9-49.4] 0.0423* (MW)

0.48 (MW)

0.18 (MW)

Gender

 Male

 Female 

69.5% (41/59)

30.5% (18/59) 

33.3%(8/24)

66.7% (16/24) 

0.0025 * (C)

0.0025 * (C)

Low BMD

 Male

 Female 13.6% (8/59)

13.6% (8/59)

0.0% (0/59) 37.5% (9/24)

20.8% (5/24)

16.7% (4/24) 0.0324 * (F)

0.51 (F)

0.0058 * (F)

Mono-infected (HIV)

 Male

 Female 81.4% (48/59)

57.6% (34/59)

23.7% (14/59) 83.3% (20/24)

25% (6/24)

58.3% (14/24) 0.83 (C)

0.007* (C)

0.003* (C)

Co-infected (HIV and HCV)

 Male

 Female 18.6% (11/59)

11.9% (7/59)

6.8% (4/59) 16.7% (4/24)

8.3% (2/24)

8.3% (2/24) 1.00 (F)

1.00 (F)

1.00 (F)

AIDS

 Male

 Female 45.8% (27/59)

37.3% (22/59)

8.5% (5/59) 83.3% (20/24)

25.0% (6/24)

58.3% (14/24) 0.0017* (C)

0.28 (C)

<0.0001 * (C)

CD4 count (cells/μl)

<200

 200-349

 ≥350 

44.1% (26/59)

50.8% (30/59)

5.1% (3/59) 

83.3% (20/24)

16.7% (4/24)

0.0% (0/24) 

0.0012* (C)

0.0043* (C)

0.55 (F)

Abnormal CD4 count 

Male

 Female 98.3% (58/59)

67.8% (40/59)

30.5% (18/59) 100% (24/24)

33.3% (8/24)

66.7% (16/24) 1.00 (F)

0.0039 * (C)

0.0024 * (C)

CD4 count (cells/μl)

 Male

 Female 210[125-279]

200 [117.6-269.6]

265 [101.9-308.1] 61 [24.5-120]

37.5 [7.5-181.9]

72.5 [29.6-141.7] 0.0003*(MW)

0.0264* (MW)

0.0016* (MW)

AbnormalBMI

 Male

Female 45.8% (27/59)

33.9% (20/59)

11.9% (7/59) 41.7% (10/24)

12.5% (3/24)

29.2% (7/24) 0.73 (C)

0.0483 * (C)

0.10 (F)

BMI

 Male

 Female 20 [19-21]

19.8 [17.8-21]

21 [19.2-23.2] 18.3 [18-19]

18.3 [17.8-20.6]

18.4 [18-19.4] 0.11 (MW)

0.80 (MW)

0.0079 * (MW)

Previous fractures

 Male

 Female 8.5% (5/59)

6.8% (4/59)

1.7% (1/59) 12.5% (3/24)

4.2% (1/24)

8.3% (2/24) 0.68 (F)

1.00 (F)

0.20 (F)

Abnormal 25-Hydroxy-vitamin D

 Male

 Female 100% (59/59)

69.5% (41/59)

31.5% (18/59) 100% (24/24)

33% (8/24)

66.7% (16/24) 1.00 (F)

0.0024* (C)

0.0024* (C)

25-Hydroxy-vitamin D (ng/mL)a

 Male

 Female 14.8±6.4

15.6±6.9

12.9±4.4 11.8±5.4

9.8±5.9

12.8±5.2 0.0466 * (T)

0.0324 * (T)

0.94 (T)

Abnormal FBD

 Male

 Female 0.0% (0/59)

0.0% (0/59)

0.0% (0/59) 0.0% (0/24)

0.0% (0/24)

0.0% (0/24) 



FBD

 Male

Female -0.2±0.8

-0.4±0.8

0.3 [-0.3;0.8] -0.03±0.9

-0.2±0.5

0.1 [-0.6;0.4] 0.37 (T)

0.43 (T)

0.31 (MW)

Abnormal LBD

 Male

 Female 13.6% (8/59)

13.6% (8/59)

0.0% (0/59) 37.5% (9/24)

20.8% (5/24)

16.7% (4/24) 0.0324 * (F)

0.51 (F)

0.0058 * (F)

LBD

 Male

 Female -0.7±1.2

-1.0 [-1.1; -0.8]

0.05±0.9 -1.2±1.4

-2.3 [-2.9; -1.3]

-0.8±1.3 0.07 (T)

0.0272 * (MW)

0.0291 * (T)

* = Significant test; T =unpaired t- test; C = chi-square test; F = Fisher’s exact test; ItG = Italian group, MiG = migrant group; Low BMD = low Bone Mineral Density ; FBD = Femoral bone density evaluated with Z-score; LBD = Lumbar bone density evaluated with Z-score; BMI = Body Mass Index; MW = Mann-Whitney test;

Discussion section 

• are there any longitudinal - possibly long-term - data from Italy?

The authors in the discussion section suggested a multicentric and prospective study 

• could that have been the case for Italian women? (osteophyte formation,

bone sclerosis, disk space narrowing, and spondylolisthesis were positively correlated with lumbar spine BMD) 

As indicated by the authors, the female gender seems to be more influenced by lifestyle independently from the ethnics. 

• So, TAF instead of TDF! or no tenofovir?

Response 

Thank you for your comment in this regard. The authors have introduced the following sentence and reference.

The findings of a high prevalence of low BMD in infected HIV migrant female and male patients aged less than 50 years of age could suggest that the first screening for osteoporosis should be carried out as soon as possible and before the start of ART. It could help the clinician in ARV management. In fact, planning HIV therapy to prevent bone loss suggesting the use of Tenofovir alafenamide (TAF) molecule instead of tenofovir reduce the risk of bone toxicity �36� A recent study enrolled young HIV positive with low BMD showed tenofovir as a molecule in ARV therapy �11�. 

36. Shafran SD, Di Perri G, Esser S, Lelievre JD, Parczewski M. Planning HIV therapy to prevent future comorbidities: patient years for tenofovir alafenamide. HIV Med 2019; 20 (Suppl 7): 1–16. 

• Did you provide osteoporosis therapy to any of your patients?

Taking into account the suggestion of the reviewer, the authors have been introduced in the following sentences:

According to osteoporosis foundation (http://share.iofbonehealth.org/WOD/Compendium/2019-IOF-Compendium-of-Osteoporosis-PRESS.pdf) the authors stress the strategies for prevention osteoporosis and the role of nutrition in maintaining bone health associated with an personalized anti HIV treatment and vitamin supplementation to reduce early bone loss baseline DXA should be performed in HIV migrants and refugees as soon as possible �13,17,20,23�and high-risk groups should be identified, and osteoporosis investigations should be carried out earlier than at 50 years of age.

Overall, the manuscript is well written. Below are a few suggestions that may improve the manuscript.

We re-organized the manuscript, including some sections that were not present in the original version, such as the limitations of this investigation. We are incredibly grateful to the reviewers for the numerous suggestions and comments that helped us improve this manuscript. Indeed, it was a lot of work, but we are very grateful for the extreme positive criticisms and comments.

The manuscript was fully revised by a native English speaker and the clean version should not contain any Grammar errors or misspellings.

---

## [Decision Letter · Decision Letter 1]

13 Jul 2020

PONE-D-20-08720R1

Low Bone Mineral Density in HIV-positive Young Italians and Migrants

PLOS ONE

Dear Dr. Sergi,

Thank you for submitting your manuscript to PLOS ONE. After careful consideration, we feel that it has merit but does not fully meet PLOS ONE’s publication criteria as it currently stands. Therefore, we invite you to submit a revised version of the manuscript that addresses the points raised during the review process.

Specifically: The authors should follow the minor comments/suggestions of the reviewer-2 as noted below proof reading the manuscript for grammatical errors and to further improve.

We look forward to receiving your revised manuscript.

Kind regards,

Dr. Sakamuri V. Reddy

Academic Editor

PLOS ONE

Reviewers' comments:

Reviewer's Responses to Questions

**Comments to the Author**

1. If the authors have adequately addressed your comments raised in a previous round of review and you feel that this manuscript is now acceptable for publication, you may indicate that here to bypass the “Comments to the Author” section, enter your conflict of interest statement in the “Confidential to Editor” section, and submit your "Accept" recommendation.

Reviewer #2: (No Response)

2. Is the manuscript technically sound, and do the data support the conclusions?

Reviewer #2: Partly

3. Has the statistical analysis been performed appropriately and rigorously? 

Reviewer #2: Yes

4. Have the authors made all data underlying the findings in their manuscript fully available?

Reviewer #2: Yes

5. Is the manuscript presented in an intelligible fashion and written in standard English?

Reviewer #2: No

6. Review Comments to the Author

Reviewer #2: General points: You went through great lengths to address most of the reviewers points- thanks!

Still, PLEASE find somebody who is proficient in English or use the F7 key in Microsoft Word to correct the many glaring grammatical mistakes in your paper. THANKS!

I stick with my opinion that your main finding, the difference in BMD between migrants and native Italians concerns only women and it may or may not have anything to do with HIV. If no data menopause status were available for your study, you should state so in Methods and mention this as a limitation in your Discussion. An alternative explanation would be differences in sun exposure and this could involve cultural factors, depending on the origin/culture/religious habits of the migrant women in your study. Table 2 with the 25-OH Vitamin D values is confusing me, particularly the "abnormal" (<12 ng/mL?) category doesn't seem to make sense - was really everybody abnormal??? Migrant women seem to have much higher levels than men, which is unexpected - the opposite was true for native Italians. PLEASE, look at your source data again and make sure that you are publishing correct numbers....

ITA MIG

Abnormal 25-OH-vitD 100% (59/59) 100% (24/24) 1.00 (F)

Male 69.5% (41/59) 33% (8/24) 0.0024* (C)

Female 31.5% (18/59) 66.7% (16/24) 0.0024* (C)

25-OH-vitD (ng/mL) 14.8±6.4 11.8±5.4 0.0466 * (T)

Male 15.6±6.9 9.8±5.9 0.0324 * (T)

Female 12.9±4.4 12.8±5.2 0.94 (T)

Finally, I like the color-coded z-score Figures that are juxtoposing the ITA and the MIG group. However, I think they ought to be c o m p l e t e l y rearranged - make one figure for femoral and one for lumbar BMD, ITA group left, MIG group right.

And then SORT the individual values for each group (maintaining stratification by gender) from lowest to highest. This will make the most meaningful graphical depiction of your findings.

For the CD4 / BMD scatterplots, I would keep the scale of the x-axis for both groups identical, i.e. considerably shorten the x-axis for the MIG group. This will make the groups easier to compare.

7. PLOS authors have the option to publish the peer review history of their article (what does this mean?). If published, this will include your full peer review and any attached files.

Reviewer #2: **Yes: **Henning Drechsler

---

## [Author Response · Author response to Decision Letter 1]

5 Aug 2020

Reviewer #2: General points: You went through great lengths to address most of the reviewers' points- thanks!

Still, PLEASE find somebody who is proficient in English or use the F7 key in Microsoft Word to correct the many glaring grammatical mistakes in your paper. THANKS!

Thank you for your comments. Yes, we revised the manuscript carefully, and one colleague with English as a native language went through it.

[Reviewer 2]: I stick with my opinion that your main finding, the difference in BMD between migrants and native Italians, concerns only women, and it may or may not have anything to do with HIV. If no data menopause status were available for your study, you should state so in Methods and mention this as a limitation in your Discussion. An alternative explanation would be differences in sun exposure, and this could involve cultural factors, depending on the origin/culture/religious habits of the migrant women in your study. 

Thank you for your comment, and we changed the manuscript appropriately. According to the menopause definition (amenorrhea for ≥12 consecutive months with symptoms suggestive of menopause and in which other causes of amenorrhea have been ruled out and/or the FSH level is elevated) our female sample showed menopause status in 11/16 (68.75%) of female migrant and 12/18 of Italian female (66%). We added this data to the manuscript.

We changed the Discussion as well.

As reported in other studies, we observed early natural menopause status both in Italian than in migrant women. This finding was analogous to both groups. Apart from HIV-related immunologic status (CD4 count and viral load) other socio-demographic variables (marital status, parity, education, or income) and religious clothing of the female migrant sample may have influenced the low BMD observed in our study. 

(Guilherme Amaral Calvet, Beatriz Gilda Jegerhorn Grinsztejn, Marcel de Souza Borges Quintana, Monica Derrico, Emilia Moreira Jalil, Andrea Cytryn, Angela Cristina Vasconcelos de Andrade, Ronaldo Ismerio Moreira, Marcelo Ribeiro Alves, Valdiléa Gonçalves Veloso dos Santos, Ruth Khalili Friedman, Predictors of early menopause in HIV-infected women: a prospective cohort study, American Journal of Obstetrics and Gynecology, Volume 212, Issue 6, 2015, Pages 765.e1-765.e13, ISSN 0002-9378, https://doi.org/10.1016/j.ajog.2014.12.040. (http://www.sciencedirect.com/science/article/pii/S0002937814024995)

[Reviewer 2]: Table 2 with the 25-OH Vitamin D values is confusing me, particularly the "abnormal" (<12 ng/mL?) category doesn't seem to make sense - was really everybody abnormal??? Migrant women seem to have much higher levels than men, which is unexpected - the opposite was true for native Italians. PLEASE, look at your source data again and make sure that you are publishing correct numbers....

ITA MIG

Abnormal 25-OH-vitD 100% (59/59) 100% (24/24) 1.00 (F)

Male 69.5% (41/59) 33% (8/24) 0.0024* (C)

Female 31.5% (18/59) 66.7% (16/24) 0.0024* (C)

25-OH-vitD (ng/mL) 14.8±6.4 11.8±5.4 0.0466 * (T)

Male 15.6±6.9 9.8±5.9 0.0324 * (T)

Female 12.9±4.4 12.8±5.2 0.94 (T)

Thank you, we reviewed all data again and made the corrections. All data have been verified and validated using a triple code identifier. All data are also available for consultation. 

Concerning the normal range of Vitamin D, it was corrected in [30-80] (ng/mL) with an analogous concentration rate in the paper. About the statistical tests, all patients in MiG and ItG had a low value of vitamin D, but in MiG was significantly lower than the ItG (T-test). Since the results were incomplete, we adopted a comparison of proportions. We tested the proportion rates between MiG and ItG for males and females:

Male (with abnormal value) 69.5% (41/59) vs. 33% (8/24) 0.0024* (C)

Female (with abnormal value) 31.5% (18/59) vs. 66.7% (16/24) 0.0024* (C)

 ItG 100% (59/59) vs. MiG 100% (24/24)

We consulted our statisticians to answer to your question.

[Reviewer 2]: Finally, I like the color-coded z-score Figures that are juxtaposing the ITA and the MIG group. However, I think they ought to be c o m p l e t e l y rearranged - make one figure for femoral and one for lumbar BMD, ITA group left, MIG group right. And then SORT the individual values for each group (maintaining stratification by gender) from lowest to highest. This will make the most meaningful graphical depiction of your findings.

Thank you for your suggestion. We agree that such figures would make the best of our data, but to obtain this type of figure, there must be groups of the same size, according to our statisticians. To represent a heatmap, a 2-column matrix with the same number of rows was used. Finally, we plot the figure in this way, but the gradualness of the colors in some parts was difficult to appreciate. Therefore, we decided to consider the order of the patients, as reported in the database. We consulted our statisticians to answer to your question and reviewed the manuscript with them.

[Reviewer 2]: For the CD4 / BMD scatter plots, I would keep the scale of the x-axis for both groups identical, i.e. considerably shorten the x-axis for the MIG group. This will make the groups easier to compare.

Thank you for your suggestion. In this case, we did consider for all scatterplots an analogous scale of the x-axis. Indeed, the scale of values on the x-axes, makes the differences evident between MiG and ItG. We consulted our statisticians to answer to your question and reviewed the manuscript with them.

---

## [Editor Report · Decision Letter 2]

7 Aug 2020

Low Bone Mineral Density in HIV-positive Young Italians and Migrants

PONE-D-20-08720R2

Dear Dr. Sergi,

We’re pleased to inform you that your manuscript has been judged scientifically suitable for publication and will be formally accepted for publication once it meets all outstanding technical requirements.

Kind regards,

Dr. Sakamuri V. Reddy

Academic Editor

PLOS ONE